# Efficacy of Neuro-Psychomotor Approach in Children Affected by Autism Spectrum Disorders: A Multicenter Study in Italian Pediatric Population

**DOI:** 10.3390/brainsci11091210

**Published:** 2021-09-14

**Authors:** Mariarosaria Caliendo, Anna Di Sessa, Elisa D’Alterio, Alessandro Frolli, Domenico Verde, Diego Iacono, Palmira Romano, Luigi Vetri, Marco Carotenuto

**Affiliations:** 1Centro di Riabilitazione “La Filanda-Lars”, 84087 Sarno, Italy; m.caliendo@hotmail.it; 2Department of Woman, Child, and General and Specialized Surgery, University of Campania “Luigi Vanvitelli”, 81100 Caserta, Italy; anna.disessa@unicampania.it; 3Clinic of Child and Adolescent Neuropsychiatry, Department of Mental Health and Physical and Preventive Medicine, University of Campania “Luigi Vanvitelli”, 81100 Caserta, Italy; elidalterio@libero.it (E.D.); marco.carotenuto@unicampania.it (M.C.); 4Disability Research Centre, University of International Studies in Rome, Via Cristoforo Colombo, 00147 Rome, Italy; alessandro.frolli@unint.eu; 5Finds-Italian Foundation for Neuroscience and Neurodevelopmental Disorders, 81040 Caserta, Italy; 6Centro di Riabilitazione “CinziaSantulli” SRL, 81031 Aversa, Italy; domenicoverde@virgilio.it; 7Neurodevelopmental Research Lab, Biomedical Research Institute of New Jersey (BRInj), Morristown, NJ 07960, USA; Iacono@brinj.org; 8Neuroscience Research, MidAtlantic Neonatology Associates (MANA), Atlantic Health System (AHS), Morristown, NJ 07927, USA; 9Neuropathology Research, MANA-BRInj, Cedar Knolls, NJ 07927, USA; 10Centro di Riabilitazione LARS, 84087 Sarno, Italy; palmi.romano@gmail.com; 11Oasi Research Institute-IRCCS, 94018 Troina, Italy

**Keywords:** autism spectrum disorders, ASDBI, neuro-psychomotor approach, therapist

## Abstract

Background: Autism Spectrum Disorder (ASD) is characterized by impairments in social interaction and reciprocal communication. ASD affects about 1% of the general population and is associated with substantial disability and economic loss. A variety of approaches to improve the core deficits and lives of people with ASD have been developed, including behavioral, developmental, educational, and medical interventions. The main objective of this study was to evaluate the efficacy of a neuro-psychomotor approach in children affected by ASD. Methods: The sample consisted of 84 children (66 males, mean age 56.9 ± 15.8 months) affected by ASD assessed between September 2020 to March 2021. The trained therapist was asked to complete the ASD behavior inventory (ASDBI) test at baseline (T0) (September 2020) and after six months (T1) (March 2021) to assess the child’s evolution over the observational period. The study was carried out in southern Italy (Campania Region). Results: ASD children showed a significant improvement for AUTISM composite after 6 months of neuro-psychomotor treatment (T1) compared to baseline (65.4 ± 12.2 vs. 75.8 ± 11.5, *p* < 0.0001). In particular, significant changes were observed for such domains as the problems of excitability (ECCIT), aggression (AGG), behaviors in social relations (RELSOC), expressive (all *p* < 0.001), sense/perceptual contact modes (SENS) (*p* = 0.0007), ritualisms/resistance to changes (RIT) (*p* = 0.0002), pragmatic/social problems (PPSOC) (*p* = 0.0009), specific fears (FEARS) (*p* = 0.01), and learning and memory (AMLR) (*p* = 0.0007). No differences for the domains Semantic/pragmatic problems (PPSEM) and language (LESP) were found. Conclusions: Our preliminary results suggest the usefulness of the neuro-psychomotor treatment in children with ASD. Although promising, these findings need to be tested further to better understand the long-term effects of this specific type of approach.

## 1. Introduction

Autism Spectrum Disorder (ASD) has been recognized as a global public health concern characterized by social communication deficits, stereotyped interests, and restricted or repetitive behaviors [1,2]. According to the fifth edition of the Diagnostic and Statistical Manual of Mental Disorders (DSM-5), it has been classified as a neurodevelopmental disorder. To date, a dramatic global increase of ASD prevalence in the last decades has been reported, with a median prevalence rate of 62 per 10,000 children [1]. Given its early onset, the complex burden of related impairments (including psychiatric comorbidities) [2,3], and the lack of current cure, there is need for early intervention programs [3,4,5]. Furthermore, the impact of ASD overwhelms the lives of the patients by also affecting those of their families and caregivers. Therefore, ASD children may take advantage of an early diagnosis and a strict management in order to improve rehabilitative outcomes [4,5,6].

The main characteristics underlying the diagnosis of ASD (such as persistent deficit of social communication and social interaction in multiple contexts, and restricted, repetitive patterns of behavior, interests or activities) may have various degrees of severity and be further aggravated by the presence of additional medical problems (including epilepsy, sleep and gastrointestinal problems, intellectual/cognitive impairment, and, in some children, presence of challenging maladaptive behaviors such as hyperactivity, anxiety, irritability, self-injury, and aggression) [2,7,8]. Effective treatment for children with ASD should involve an interdisciplinary approach, which should include a combination of educational interventions, psychological/behavioral therapies, speech-language therapy, occupational/physical therapy and medical treatments (e.g., psychopharmacology) [8,9].

Given the complexity of this condition, in which medical, psychological, social, educational and even ethical and existential aspects need to be carefully balanced, different diagnostic and rehabilitative approaches have been proposed [7,8,9]. However, there is no consensus regarding this variety of treatments [10,11,12,13].

Among ASD therapeutic options, the neuro-psychomotor treatment, commonly used in Italy, is part of these approaches. It has been referred to as psychomotricity in other European countries and it is also known as play therapy [14,15,16,17]. In the United States, a similar treatment may be represented in developmental therapy [18].

Commonly, the psychomotor approach is an expression/denomination used to indicate a European approach for education, prevention and therapy, which was developed by Aucouturier and Lapierre [19]. This approach focuses on the development of how children process information through movement during play sessions. Specifically, it proposes the following objectives: to favor the appearance of social signalers (eye contact, reference look, smile, etc.), to increase attention times, to facilitate a more appropriate use of objects, to stimulate communication, to enrich vocabulary, and to discourage certain behaviors (hyperactivity, motor stereotypes, self-injurious behaviors, etc.) [20]. In this framework, playing has gained remarkable therapeutic significance by representing a tool that enhances in a natural way the cognitive functioning of children with ASD [16,17].

This therapy represents an intervention that fits into the child’s potential development area to support and strengthen his functional areas, and requires a specific mental setting that is based on a global approach to the child, considered in its uniqueness and global aspects [21]. All children affected by ASD have interactive styles, functional, psychomotor and cognitive characteristics; the neuro-psychomotor intervention should therefore respond to these diversified needs through a highly individualized approach. The intervention supports the child’s spontaneous action and play in its various forms, which are the essential frames that give meaning to the child’s experience. Furthermore, psychomotricity may be considered as a safe and efficacious therapy for many neurodevelopmental disorders [22].

To the best of our knowledge, evidence on the efficacy and feasibility of neuro-psychomotor therapy for children affected by ASD is lacking.

The aim of the present study was to verify the effect of six-months of neuro-psychomotor treatment in ASD children and its potential impact on different aspects of life.

## 2. Materials and Methods

For this study we decided to recruit eighty-four children from southern Italy (Campania Region) who had received a level 3 diagnosis of ASD, aged between two and nine years. The study was conducted between September 2020 and March 2021. Participants were selected according to the convenience sampling method and recruited from Rehabilitation centers in the Campania Region in Italy. This research received approval from the Ethical Committee (Protocol code 13883). The data were collected anonymously after the signing of the informed consent by the parents. The reported investigation has been carried out in accordance with the principles of the Declaration of Helsinki [23].

The inclusion criteria were: (a) have a diagnosis of ASD level 3 according to the criteria of DSM-5 [22], assessed by the administration of the Autism Diagnostic Observation Schedule (Second Edition) (ADOS 2-module 1) [24] for children, and have the Autism Diagnostic Interview-Revised (ADI-R) [2003] [25] administered to parents by qualified psychologists; (b) to have followed neuro-psychomotor therapy during the observation period; (c) be between two and nine years of age; and (d) be absent of comorbid psychiatric or neurological pathologies.

Considering that the enrolled patients were diagnosed in different clinics and the extensive training needed for ADOS-2, in our clinical practice we also used the Childhood Autism Rating Scale test (CARS 2) with the aim of improving the diagnosis of ASD [26,27]. It represents one of the most reliable tools available, and it was specifically developed to rapidly identify ASD children from two years of age. The test is divided into fifteen items relating to the main areas of behavior, with scores ranging from 1 to 4 [28,29]. The sum of all scores gives an overall value with the following meanings: (1) from 15 to 30, non-autistic; (2) from 30 to 37, light to medium autism; (3) from 37 to 60, severe autism [29].

Specifically, in our sample the average scores obtained by administering ADOS 2 and CARS 2 were 20.3 (SD = 0.40) and 42 (SD = 5.00), respectively.The exclusion criteria were: (a) having followed a behavioral qualification/rehabilitation treatment; (b) the presence of sensory disturbances (neurosensory hypoacusia, blindness-hypovision); and (c) the presence of severe neurological disorders (drug-resistant epilepsy, cerebral palsy, neuromuscular diseases, etc.).

All of the enrolled ASD children received a neuro-psychomotor treatment for six months, and they were evaluated at baseline (T0-before starting treatment) and at the end of the observational period (T1-after performing the treatment). The treatment was administered by trained child therapists in outpatient settings two to five times per week. All of the therapists shared the same protocol and every child was followed by the same therapist over the six months. The standard psychomotor session length was 45 min [30].

The therapist was asked to complete the ASD behavior inventory (ASDBI) (as the Italian version of the Pervasive Developmental Disorder Behavior Inventory (PDDBI) [31,32] at baseline (T0) and after six months (T1) to assess the child’s evolution over this period.

The ASDBI is a standardized rating scale designed for parents and teachers which allows one to make an assessment of relational behavior and some other symptomatic aspects typical of children with ASD, referring to different life contexts [2]. It is a sensitive and useful evaluation tool for multiple fields (clinical, medical, educational, and research applications) [33]. Materials required for administration include the ASDBI assessment protocol and a pencil. The standard protocol was designed to be completed in 30–45 min, while its short version can be completed in 20–30 min. Standard protocol was adopted in this study because of its wider informativeness and completeness.

To include patients in the study, the therapist was asked to complete all of the items of the protocol. The ASDBI consists of 10 domains and 180 items. The domains included in the ASDBI were selected based on their relevance to the diagnosis of ASD and the behaviors most frequently associated with these disorders.

The domains are summarized in Table 1.

The rater had a Likert scale to indicate how descriptive a given attribute is of the child’s behavior. Ratings range from 0 (NEVER) to 1 (RARELY) to 2 (SOMETIMES/PARTIALLY) to 3 (OFTEN). In addition, the evaluator had the opportunity to use “?” as an option available for questions without an identifiable response.

The total score of each domain (obtained by adding the items of the various clusters) was then converted in order to obtain the T points. The composite score for autism was obtained by adding the t-scores of the SENS, RIT, PPSOC and PPSEM domains. The sum of RELSOC and LESP was subtracted from the total obtained.

The score obtained was converted into a t-score using the conversion table in the manual. People with autism got higher scores. Conversely, people having social and communication skills that do not exhibit non-adaptive behaviors had lower scores. A score of 50 ± 10 (T-points between 40 and 60) is characteristic of 68% of autism cases.

The data analysis was performed using the commercially available STATISTICA 8.0 package for Windows (StatSoft, Inc., Tulsa, OK, USA). Firstly, descriptive statistics through chi-squared tests were computed for demographic data. Afterwards, group comparisons were made using one-way analysis of variance (ANOVA). Statistical significance was set at *p* < 0.05.

## 3. Results

The mean age of the study population was 56.9 ± 15.8 months. Out of 84 patients, 66 were male (78%) and 49 were verbal (58%). Neuro-psychomotor therapy was administered two to five times per week (3.6 ± 0.7 h for week).

The sample, based on the suggestions of the DSM 5, was divided into three main levels based on the severity of symptoms. Level 1 (9.5%) consisted of eight children (of which 6 were males). Level 2 (34.5%) and 3 (56%) included 29 (of which 23 males) and 47 children (of which 38 males), respectively.

At baseline (T0), the CARS2 score was distributed as follows: severe symptoms of autistic spectrum disorder (>37): 65; moderate spectrum disorder symptoms (30–36.5): 18; minimum signs or no symptoms of spectrum disorder (15–29.5): 1.

After six months (T1), the CARS2 score distribution was as follows: severe symptoms of autistic spectrum disorder (>37): 55; moderate spectrum disorder symptoms (30–36.5): 28; minimum signs or no symptoms of spectrum disorder (15–29.5): 1.

After six months of neuro-psychomotor treatment (T1), ASD children showed a significant reduction in the domain problems of excitability (ECCIT); aggression (AGG); behaviors in social relations (RELSOC), expressiveness (all *p* < 0.001), sense/perceptual contact modes (SENS) (*p* = 0.0007); ritualisms/resistance to changes (RIT) (*p* = 0.0002); pragmatic/social problems (PPSOC) (*p* = 0.0009); specific fears (FEARS) (*p* = 0.01), learning and memory (AMLR) (*p* = 0.0007) (Table 2).

Changes observed in Semantic/pragmatic problems (PPSEM), and language (LESP) domains were not statistically significant (*p* = 0.63 and *p* = 0.204, respectively) (Table 2).

Composite scores were also calculated. A significant improvement was found for RIPRIT/C (obtained by adding Sense/perceptual contact modes, ritualisms/resistance to changes, pragmatic/social problems and semantic/pragmatic problems and converting into points T) (*p* < 0.0001), PCI/C (obtained by summing the first seven domains) (*p* < 0.0001), AECS/C (obtained by summing RELSOC and LESP) (*p* = 0.002) and AUTISM composite (*p* < 0.0001) (Table 3).

## 4. Discussion

Our study first investigated the effect of a six-month neuro-psychomotor treatment in ASD children by demonstrating significant improvements in the areas mainly compromised in these patients, in particular those relating to social communication, mutual social interaction and functional and symbolic play.

Neuro-psychomotor treatment has been proposed as a valid therapeutic option in several neuropsychiatric disorders such as separation anxiety, selective mutism, maltreatment, elimination disorders, and physical abuse [22,29]. It is well established in some European countries (including Italy and France) and it considers body experience as a fundamental element in the development of a person’s identity and as an expression of emotional life and the evolution of cognitive processes. Working on the body provides the pleasure of self-unity within a kinesthetic and sensorial container which allows the child to overcome hesitations, tonic-emotional resistances, and anxieties.

Autism is characterized by rigid and repetitive behaviors. ASD patients love to implement their repetitive behaviors because these give them security and predictability in a way that is perceived as chaotic and unpredictable. Notably, altered motor performance has been considered as a potential early sign of ASD [34,35]. In this framework, previous reports have highlighted the helpful role of a neuro-psychomotor assessment for the critical identification of ASD children at an early stage, by expanding also knowledge on developmental organization of these patients [36,37].

According to the neuro-psychomotor approach, therapists insert “gradually” into these rituals, making changes in all contexts of the child’s life and trying to extinguish a behavioral problem by teaching a replacement and reinforcing this later at each time of occurrence.

Neuro-psychomotor intervention also favors the emergence of social interaction skills through role-playing, but also through social narratives, which can help people with autism develop greater social understanding. Among specific aims of this therapeutic approach, improvements to the child’s environmental strategies are included. For instance, any causes of sensory hypersensitivity, which could be the cause of the child’s arousal, are sought. Therapists also implement an effective education of emotional regulation, which is not achieved by “preaching” emotional or motor self-control to the child, but rather by “offering” containment through an empathic sharing of his emotional experience.

In addition, rates of aggressive behavior may be higher in individuals with ASD compared to their typically developing peers and those with other developmental disabilities [38]. Aggression is clearly associated with negative outcomes for children with ASD, including impaired social relationships [38]. Understandably, addressing aggressive behavior is pivotal to improving outcomes for individuals with ASD and their caregivers [39]. As a novel perspective, our findings seem to support a global significant improvement in specific traits (such as ECCIT AGG, RELSOC, SENS, RIT, PPSOC, FEARS, and AMLR domains) after six months of neuro-psychomotor treatment in ASD children, except for the language domain (including PPSEM and LESP), which likely requires not only a more specific approach but also a longer duration of therapy. This could be due to the natural indirect action of the neuro-psychomotor approach on language and pragmatic skills.

However, we are aware that the study has some limitations that should be mentioned. The sample size of the study was limited and the longitudinal evaluation of long-term outcomes is lacking. More, the ASDBI represented an operator dependent test.

Despite these limitations, our preliminary results suggest a positive effect of the psychomotor approach in ASD children. Greater validation is needed to better elucidate the role of this therapeutic approach in the ASD context.

## 5. Conclusions

ASD represents a tangled neurodevelopmental disorder that is mainly characterized by social dysfunction. It usually affects both the affective and cognitive dimensions of the patient with a serious burden of different impairments that might also be considered extremely disabling to parents and their lives. Behavioral treatments have been recommended as treatments for ASD. In this perspective, the neuro-psychomotor approach represents a therapeutic option that is non-intrusive for families, relatively inexpensive, adaptable to the clinic or home, and to groups or individuals. In particular, this therapeutic approach can be considered as tailored for children, leading to a great adaptability that fits well to the particular features of the disorder.

As observed by Aucouturier et al. [17], “the child is action, play and emotion” and all of these features are protected and strengthened in neuro-psychomotor therapy.

## Figures and Tables

**Table 1 brainsci-11-01210-t001:** Description of ASDBI domains [33].

** SENS **	Sense/perceptual contact modes	Visual behaviors; tactile or olfactory processing of inedible objects; behaviors that produce noise; proprioceptive/kinesthetic behaviors; repetitive manipulative behaviors; kinesthetic behaviors and balance
** RIT **	Ritualisms/resistance to changes	Resistance to changes in the environment; resistance to changes in planned/routine activities; rituals.
** PPSOC **	Pragmatic/Social Problems	Social contact problems; social awareness problems; inappropriate relationships.
** PPSEM **	Semantic/pragmatic problems	Speech abnormalities; problems in understanding words; verbal-pragmatic deficits.
** ECCIT **	Problems of excitability	Kinesthetic behaviors and reduced responsiveness
** FEARS **	Specific fears	Behavior of social isolation; avoidance of sounds; excessive or unusual fear and anxiety.
** AGG **	Aggression	Self-aggressive behaviors; incongruous negative reactions; problems when people who care for him/her or other significant figures return from a vacation or return to school after an absence; aggression towards others; general problems of character.
** RELSOC **	Behaviors in Social Relations	Visual behaviors in social relationships; positive affective behaviours; gestures in social relationships; reactions to signals that invite to interrupt or inhibit inappropriate social behaviors; social play behaviors; symbolic play behaviors; empathic behaviors; social imitation behaviors
** LESP **	Expressive language	Vocal productions; productions of consonants at the beginning, middle and end of the word; production of diphthongs; skills in expressive language; emotional verbal intonation; pragmatic skills in conversation.
** AMLR **	Learning and memory	Memory skills, receptive language, association-based skills.

**Table 2 brainsci-11-01210-t002:** Description of the ASDBI scale scores in ASD children affected at baseline (T0) and after six months (T1) of neuro-psychomotor therapy. Improvements for the first seven domains are expressed as decreased scores, while for the RELSOC, LESP and AMLR domains, as increased scores.

	ASD T0 (*n* = 84)	ASD T1 (*n* = 84)	* p *
SENS	66.44 ± 12.68	59.85 ± 11.99	0.0007 *
RIT	69.74 ± 15.09	61.59 ± 13.46	0.0002 *
PPSOC	66.38 ± 11.29	60.38 ± 11.30	0.0009 *
PPSEM	55.39 ± 17.04	54.91 ± 15.39	0.63
ECCIT	58.27 ± 7.48	51.48 ± 8.25	<0.0001 *
PAURE	59.93 ± 12.61	55.32 ± 11.51	0.01 *
AGG	66.58 ± 16.10	59.19 ± 14.43	<0.0001 *
RELSOC	35.80 ± 11.29	43.28 ± 11.12	<0.0001 *
LESP	40.76 ± 10.12	42.95 ± 10.56	0.204
AMLR	42.43 ± 11.02	48.66 ± 11.74	0.0007 *

* *p*-value statistically significant.

**Table 3 brainsci-11-01210-t003:** Description of the ASDBI scale composite score among children affected by ASD at baseline (T0) and after six months (T1) of neuro-psychomotor therapy.

	ASD T0 (*n* = 84)	ASD T1 (*n* = 84)	* p *
RIPRIT/C	72.39 ± 14.30	63.66 ± 13.51	<0.0001 *
PCI/C	71.16 ± 14.10	62.02 ± 13.77	<0.0001 *
AECS/C	37.04 ± 10.91	42.19 ± 10.63	0.002 *
AUTISM	75.85 ± 11.56	65.41 ± 12.21	<0.0001 *

* *p*-value statistically significant.

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
