# Peer review of "Efficacy of Neuro-Psychomotor Approach in Children Affected by Autism Spectrum Disorders: A Multicenter Study in Italian Pediatric Population"

_brainsci, 2021, doi:10.3390/brainsci11091210_

Round 1

Reviewer 1 Report

This study tries to echo the impacts of a psychomotor intervention on the core symptoms of autism spectrum disorders (ASD).

By using a scale ASDBI the impacts of the intervention after 6 months are presented.  The authors are trying to introduce this approach as an effective intervention. Hence the adopted approach and presented data might not echo the real impacts of this intervention.

However, I have major methodological reservations about the data collected.

First, I have great skepticism about the adopted methodology. Using quasi-experimental research designs might not support neuro-psychomotor therapy as an approved and well-supported approach for the diverse needs of individuals with ASD. I tried to check the presented references and none of them were supporting the applicability of this approach on individuals with ASD and also implying ply therapy to is approach might be not very helpful since play is a modality for all other interventions for different types of developmental disabilities.

More supporting references are needed to support this claim especially when it is said that this approach is applicable in Europe, therefore more supporting references are needed.

As a certified ASD evaluator, Is not understand the application of 3 measures for the diagnosis of one condition. I should notify that ADOS2 consisted of 5 different modules (one for the preschoolers and 4 for different ages and language abilities). ADOS2 application is time-consuming and when you have done this what was the reason for applying CARS2 as well? And you checked all the recruited samples once again with the DSM5 criteria? Following important points should be clarified:

  1. why this approach is considered (using 3 scales)
  2. what was the level of overlap and difference between these scales for each child?
  3. since the age range of the recruited sample was 2 to 9 you should decide about the applicable ADOS2 modules, so, which modules applied?
  4. also, data regarding the Italian version of the scales and their validity and reliabilities are needed.

I would like to see an analysis of the two scales (ADOS2 and CARS 2) items together to see if any redundant items partly explain the reported significant changes.

A second application might indicate if the sample still receives the diagnosis of ASD or not but I do not request 2nd round application of the diagnostic scales.

But the most important issue is considering autism spectrum disorder as a single diagnosis. ASD implies a very heterogeneous group of individuals with different levels of abilities and disabilities. Using different scales, it should be easy to categorize the sample based on DSM5 suggestion into 3 main levels according to the severity of the symptoms.

Finally, two little remarks: 1) How did the impacts of the other factors be controlled to understand the real impact of the independent variable? I mean as it is mentioned children were in a clinic, they of course received other types of non-neuro-psychomotor therapy protocol interventions (if it has a clear protocol for service provision) therefore, how it can be proved that the reported significant change was because of the neuro-psychomotor therapy intervention? 2)It seems to me that the impacts of this innervation might be different with different age groups and splitting the sample into two age groups might produce different findings.

In sum, this is very odd that regardless all the methodological shortcoming and shortage of the presented data regarding the sample and their other qualifications, they are not mentioned in the limitations section of this study and it seems that these issues are neglected by the authors.

The purpose of considering a control group is to allow the researcher to conclude that any change observed in the “active treatment group who received neuro-psychomotor therapy” might be due to the treatment being investigated, rather than to other factors which might not be controlled or are out of control. Having a control group is particularly important when factors in addition to the intervention under study can affect the outcome (which are 10 qualifications which are reported here), especially when the new or less-established approaches are presented. Failure to use a control group or an inappropriate control group can make it impossible to draw meaningful conclusions from a study.

There were some problems with the presented tables and from my point of view, it does not add any information and might be deleted. 

Author Response

We thank the reviewers for the time spent in reviewing our manuscript and for the opportunity to resubmit it after major revision. We revised the manuscript according to the reviewer’s comments. We provided a point-by-point response to the reviewer’s comments.

Please find attached in the word file how we have adressed your concerns.

Reviewer 2 Report

Main message of the article

The article entitled “Efficacy of neuro-psychomotor approach in children affected by autism spectrum disorders: a multicenter study in Italian pediatric population” by Caliendo et al. explores the benefits of the neuro-psychomotor approach in regard to the symptoms ASD-related. In particular, the authors evaluated the differences on 84 Italian participants after a 6-months psychomotor training and the results seem promising for the short-term understanding of such approach.

General Judgment Comments

Although the idea behind the study is interesting, the manuscript is often disorganized in its writing and not clear for the readers. Generally, the design and methods are appropriate, and the screening of symptoms of autism is well-conducted. The title summarizes well the content of the paper, but the abstract needs more work to be understandable at the first read. The statistical analyses lack of rigor. In particular, the authors are encouraged to test the ANOVA assumptions before applying it or to adopt another test in order to assess the outcomes of the psychomotor training. Also, Bonferroni’s correction should be adopted to correct the significance level in relation to the number of tests that were conducted. To determine whether the sample size is appropriate, the statistical power should be computed as well. Finally, the format in which results are reported is inconsistent and needs to be revised.

I recommend Major Revision for the current article because the statistical approach needs some adjustments, and the manuscript requires some work to facilitate the reading.

Major Issues

  • The Introduction section needs to be better organized with a clearer structure that would facilitate the reading. To do so, Lines 52-55 could be moved at the initial part of the Introduction in order to provide a definition of autism for the readers. Also, adding few other details about the way in which the psychomotor approach is implemented and conducted could be helpful to understand the approach.
  • Why was one of the inclusion criteria set at being 2-9 years old? Can the authors please provide a justification?
  • Please include the psychometric properties of the scales composing the ASDBI.
  • Lines 153-155: Was the procedure to obtain SENS, RIT, PPSOC, and PPSEM validated in the literature? If yes, please provide references.
  • The authors stated that they used Chi-squared test for the demographic data. Nevertheless, it is not clear what they meant, which demographic data they considered, the hypotheses that were tested and the results of such tests.
  • The group comparison using ANOVA is not clear. Which groups were compared? Why did the authors used the ANOVA and not Student’s t-test? Also, were the ANCOVA assumptions tested before adopting such test? Was also the power computed?
  • The significance level set at 0.05 should be revised and corrected for the number of tests that were conducted.
  • Results are not reported in a consistent and standard way. Please revise.
  • Which scale is “expressive”? In the results, the total scores for ASDBI emerge to be 11 if we include “expressive”.
  • Please fix the Informed Consent Statement.

Minor Issues

  • Line 26: Rather than “The results involve…”, I would say “The sample consists of …”
  • Line 28: The extended name for ASDBI should be specified.
  • In the Abstract, it would be good to specify that the research was conducted in Italy.
  • The authors states throughout the paper that ASD symptoms were assessed in two timepoints, one in September and the other in April. Nevertheless, this timespan corresponds to 7 months and not 6, as the authors stated.
  • In the Abstract, sometimes the authors reported p-values and abbreviations for ASDBI scores and sometimes they did not. Please make the abstract consistent in its form.
  • Lines 44-46: What was the period of this increased in rates of autism? One year, one decade, or others? Please, also provide references.
  • Line 47: “psychosocial interventions highlight” --> “psychosocial interventions, highlight”
  • Lines 46-49: the sentence is not very clear, please rephrase
  • References format throughout the paper needs to be consistent. Sometimes references are reported between squared or round brackets (e.g., in Lines 54-55 and in Lines 59-60). Please check the references format throughout the manuscript.
  • Line 90: “strengthen his functional” --> “strengthen their functional”
  • Line 98: “The aims of the present study were” --> “The aim of the present study was”
  • Line 102: Please specify Campania Region’s country for readers.
  • Line 119: what is meant for “frequency of neuro-psychomotor therapy”?
  • Line 123: specify the extended name for “cp”
  • Line 129: specify the extended name for “ASDBI”
  • Lines 139-141: What is meant for “the therapist was asked to complete each question of the protocol, under penalty of exclusion within the study”?
  • Table 1: Please revise the caption to make the table self-explanatory. Also, the first row with titles for the reported column is missing from the table.
  • Line 150: “shave”?
  • Lines 174-175: check the approximation of numbers. The approximation is wrong for the group’s mean age and the SD for the hours of therapy.
  • Table 3: “mounths” --> “months”
  • Table 3: “frequency neuro-psychomotor therapy” --> “frequency neuro-psychomotor therapy (hours)”
  • Table 4: Please add the stars for significance levels next to the p-values.
  • Line 211: “latteras” --> “latter as”
  • Lines 217-219 are not clear. Please revise.
  • Line 211: “way”?
  • Lines 219-221: This seems a generalization. Please add references to support what stated.
  • Lines 248-250: As before, this seems a generalization. Please add references to support what stated.
  • Line 252: “there for” --> “therefore”

Final comments

I recommend Major Revision for the statistical analysis needs to be revised and more rigorous. Also, the manuscript needs more work to facilitate the reading, for often times it resulted inconsistent in its format.

Author Response

We thank the reviewers for the time spent in reviewing our manuscript and for the opportunity to resubmit it after major revision. We revised the manuscript according to the reviewer’s comments. We provided a point-by-point response to the reviewer’s comments.

Changes have been highlighted in yellow in the revised version of the manuscript.

Major Issues

-The Introduction section needs to be better organized with a clearer structure that would facilitate the reading. To do so, Lines 52-55 could be moved at the initial part of the Introduction in order to provide a definition of autism for the readers. Also, adding few other details about the way in which the psychomotor approach is implemented and conducted could be helpful to understand the approach.

Answer: thank you for your suggestion. We revised the introduction accordingly and we explained in more depth the usefulness of the psychomotor approach in this field. Please see lines 46-94 of the revised version of the text.

-Why was one of the inclusion criteria set at being 2-9 years old? Can the authors please provide a justification?

Answer: according to national rules, psychomotor treatment in Italy can be prescribed in children up to 9 years. Given that, our study population was aged 2-9 years old.

-Please include the psychometric properties of the scales composing the ASDBI.

Answer: The ASDBI test  (Emberti Benassi)represents the Italian version of the PPDBI scale of Cohen et al.,  that represents a largely validated and reliable rating scale for ASD children assessment. [Cohen IL, Schmidt-Lackner S, Romanczyk R, Sudhalter V. The PDD Behavior Inventory: a rating scale for assessing response to intervention in children with pervasive developmental disorder. J Autism Dev Disord. 2003 Feb;33(1):31-45. doi: 10.1023/a:1022226403878. PMID: 12708578.; Cohen, I.L.; Sudhalter, V. The PDD Behavior Inventory. Lutz, FL: Psychological Assessment Resources. Inc..(Manual ed.) 2005].

-Lines 153-155: Was the procedure to obtain SENS, RIT, PPSOC, and PPSEM validated in the literature? If yes, please provide references.

Answer: The procedure to obtain these domains were largely validated. Please see references 27,28 of the revised version of the text [Cohen IL, Schmidt-Lackner S, Romanczyk R, Sudhalter V. The PDD Behavior Inventory: a rating scale for assessing response to intervention in children with pervasive developmental disorder. J Autism Dev Disord. 2003 Feb;33(1):31-45. doi: 10.1023/a:1022226403878. PMID: 12708578.; Cohen, I.L.; Sudhalter, V. The PDD Behavior Inventory. Lutz, FL: Psychological Assessment Resources. Inc..(Manual ed.) 2005].

Minor Issues

-Line 26: Rather than “The results involve…”, I would say “The sample consists of …”

Answer:  we modified it accordingly.

-Line 28: The extended name for ASDBI should be specified.

Answer: following your suggestion, we specified it.

-In the Abstract, it would be good to specify that the research was conducted in Italy.

Answer: we specified it in the revised version of the abstract.

-The authors states throughout the paper that ASD symptoms were assessed in two timepoints, one in September and the other in April. Nevertheless, this timespan corresponds to 7 months and not 6, as the authors stated.

Answer: The study was carried out from September 2020 to March 2021. each patient was treated for six months and evaluated at baseline (T0) and after six months of neuro-psychomotor therapy (T1). In the previous version of the manuscript, there was a mistake. We apologies for this inconvenience and we corrected it.

-In the Abstract, sometimes the authors reported p-values and abbreviations for ASDBI scores and sometimes they did not. Please make the abstract consistent in its form.

Answer: following your suggestion, we modified it. Please see the revised version of the abstract.

-Lines 44-46: What was the period of this increased in rates of autism? One year, one decade, or others? Please, also provide references.

Answer: we specified the period of the increased rate of autism and we provided adequate reference. Please see lines 50-51 and ref 1 of the revised version of the manuscript.

-Line 47: “psychosocial interventions highlight” --> “psychosocial interventions, highlight”

Answer: we corrected it.

-Lines 46-49: the sentence is not very clear, please rephrase

References format throughout the paper needs to be consistent. Sometimes references are reported between squared or round brackets (e.g., in Lines 54-55 and in Lines 59-60). Please check the references format throughout the manuscript.

Answer: after a careful check, we formatted the references accordingly.

-Line 90: “strengthen his functional” --> “strengthen their functional”

Answer: we corrected it

-Line 98: “The aims of the present study were” --> “The aim of the present study was”

Answer: we corrected it

-Line 102: Please specify Campania Region’s country for readers.

Answer: following your suggestion, we specified both in the revised version of the abstract and of the main text.

-Line 119: what is meant for “frequency of neuro-psychomotor therapy”?

Answer: the frequency refers to hours per week of therapy.

-Line 123: specify the extended name for “cp”

Answer: following your suggestion, we specified the acronym CP (cerebral palsy) in the revised version of the manuscript.

-Line 129: specify the extended name for “ASDBI”

Answer:  following your suggestion, we specified the acronym ASDBI (Autism Spectrum Disorder Behavioral Interventions ) in the revised version of the manuscript.

-Lines 139-141: What is meant for “the therapist was asked to complete each question of the protocol, under penalty of exclusion within the study”?

Answer:  we explained this in a clearer manner in the revised version of the manuscript. In order to be included in the study, all the items of the protocol needed to be completed. Please see lines 140-141 of the revised version of the manuscript.

-Table 1: Please revise the caption to make the table self-explanatory. Also, the first row with titles for the reported column is missing from the table.

Answer:  we modified the Table 1 according to your comments. Please see the revised version of Table 1.

-Line 150: “shave”?

Answer:  we corrected this typo, thank you.

-Lines 174-175: check the approximation of numbers. The approximation is wrong for the group’s mean age and the SD for the hours of therapy.

Answer: we corrected the approximation of numbers according to your comment. Please see….

-Table 3: “mounths” --> “months”

Answer: we corrected the typo, thank you.

-Table 3: “frequency neuro-psychomotor therapy” --> “frequency neuro-psychomotor therapy (hours)”

Answer: we modified the sentence according to your suggestion.

-Table 4: Please add the stars for significance levels next to the p-values.

Answer: following your suggestion, we added the stars for significance levels next to the p-values. Please see the revised version of all the Tables.

-Line 211: “latteras” --> “latter as”

Answer: we corrected the typo, thank you.

-Lines 217-219 are not clear. Please revise.

Answer:  following your comment, we revised this sentence.

-Line 211: “way”?

Answer:  we corrected this typos, thank you.

-Lines 219-221: This seems a generalization. Please add references to support what stated.

Answer:  following your comment, we added references supporting our statement. 

-Lines 248-250: As before, this seems a generalization. Please add references to support what stated.

Answer: following your comment, we added references supporting our statement. 

-Line 252: “there for” --> “therefore”

Answer: we corrected it, thank you.

Round 2

Reviewer 1 Report

Unfortunately, I did not find the updated version of an improved submission of the previously submitted manuscript.  I understand that the authors have tried to address the issue hence, most of the serious methodological issues remained unchanged and as it was in the previous version. Instead of giving the reason for the application of the scales that may have conflicts with each other, the authors explained the scales. I asked for the discrepancies or consistencies of the applied scales but it is neglected or the reply is not satisfactory. As an example "CARS 2" is mentioned as a screening scale while it is a reliable diagnostic one. Similar neglection is done regarding the question about the applied modules of ADOS 2.  I also asked for the data on the reliability and validity of a possible Italian version of CARS, ADOS2, and other applied scales. The language in which non-Italian scales were used. These issues are all remained unanswered. 

Author Response

We thank the reviewers for their valuable suggestions and hope we have adequately addressed their previous raised issues.

Reviewer 

Unfortunately, I did not find the updated version of an improved submission of the previously submitted manuscript.  I understand that the authors have tried to address the issue hence, most of the serious methodological issues remained unchanged and as it was in the previous version. Instead of giving the reason for the application of the scales that may have conflicts with each other, the authors explained the scales. I asked for the discrepancies or consistencies of the applied scales but it is neglected or the reply is not satisfactory. As an example "CARS 2" is mentioned as a screening scale while it is a reliable diagnostic one. Similar neglection is done regarding the question about the applied modules of ADOS 2.  I also asked for the data on the reliability and validity of a possible Italian version of CARS, ADOS2, and other applied scales. The language in which non-Italian scales were used. These issues are all remained unanswered

Answer: we are very sorry for your comments after the resubmission of the manuscript. Indeed, we made great efforts to try to address the previous raised issues. However, we thank you to give us the opportunity to provide further explanation about our study at this time.

To date, CARS-2 represents one of the most reliable diagnostic scale for ASD.  About the Cronbach’s alpha, a recent systematic review and meta-analysis (Moon SJ, Hwang JS, Shin AL, Kim JY, Bae SM, Sheehy-Knight J, Kim JW. Accuracy of the Childhood Autism Rating Scale: a systematic review and meta-analysis. Dev Med Child Neurol. 2019 Sep;61(9):1030-1038. doi: 10.1111/dmcn.14246. Epub 2019 Apr 11. PMID: 30977125) reported a value within the range of 0.82 to 0.95 (higher than the generally acceptable range 0.6 or 0.7). More, the sensitivity was considered as acceptable, while the specificity was not, by suggesting the use of CARS-2 along with other confirmatory tools. Despite the long administration and the required extensive training needed for ADOS-2 (considered as the diagnostic gold standard for ASD), CARS-2 provides a rapid acquisition of quantitative ratings of ASD severity through a direct observation (Sanchez MJ, Constantino JN. Expediting clinician assessment in the diagnosis of autism spectrum disorder. Dev Med Child Neurol. 2020 Jul;62(7):806-812. doi:10.1111/dmcn.14530. Epub 2020 Apr 2. PMID: 32239502; PMCID: PMC7540056). Thus, when used in addition to others diagnostic tools, CARS-2 might help to avoid diagnostic delays.

Therefore, in our study, considering also the features of the enrolled patients (diagnosed in different Clinics- not all in our Department- and then recruited from various Rehabilitation Centers of Campania Region in Italy), we used both scales.

More, we have improved the description of the sample and of the tools used. Specifically, we have clarified the form adopted during the administration of ADOS 2 (used for diagnosis). We entered the average scores obtained from the tests administered (ADOS 2 and CARS 2). Furthermore, with regard to the information on the validity and reliability of the diagnostic tests used (ADOS 2 and CARS 2) we add that we have used the American version of the tests and its relative standardization, highly valid and reliable and used as a clinical and diagnostic tool in Italy as they are the only existing versions. There are no Italian standardizations. In addition, the diagnostic evaluation was performed by referring to the diagnostic criteria of DSM 5, an American psychiatric manual, with a very high validity and reliability, also used in Italian clinical practice by most.

We add that we have decided not to calculate Cronbach's alpha as the sample we have recruited is small, and therefore we have referred to the American standardization of the diagnostic tests used and its validity and reliability data. Moreover, these tools (ADOS 2 and CARS 2) were used to make a diagnosis in order to recruit the subjects in the examined sample, but the goal of our study was precisely to evaluate the effectiveness of the treatment. To verify our research hypothesis, it was not necessary to use these tools, so calculating the validity and reliability indices was not necessary for the purposes of our study.                                                                                                 

Regarding the reliability and validity of other scales, we previously attempted to add these data in the revised version of the manuscript. ASDBI (Emberti Benassi) represents the Italian version of the PDDBI of Cohen et al. [Cohen IL, Schmidt-Lackner S, Romanczyk R, Sudhalter V. The PDD Behavior Inventory: a rating scale for assessing response to intervention in children with pervasive developmental disorder. J Autism Dev Disord. 2003 Feb;33(1):31-45. doi: 10.1023/a:1022226403878. PMID: 12708578.; Cohen, I.L.; Sudhalter, V. The PDD Behavior Inventory. Lutz, FL: Psychological Assessment Resources. Inc..(Manual ed.) 2005]. It is a widely reliable and validated rating scale designed to assess Pervasive Developmental Disorder in children.

Please see the material and methods section of the revised version of the manuscript.

Reviewer 2 Report

The article is almost ready, as minor edits, I suggest adding more references on the neuro-psychomotor aspects involved in ASD so the reader gets a better understanding of why is needed the treatment. For example, this article published on a mdpi journal is relevant:

https://www.mdpi.com/2073-8994/1/2/215/pdf

It will be nice to see this article published as it will be an extension of the literature

Author Response

We thank the reviewers for their valuable suggestions and hope we have adequately addressed their previous raised issues.

Reviewer 

The article is almost ready, as minor edits, I suggest adding more references on the neuro-psychomotor aspects involved in ASD so the reader gets a better understanding of why is needed the treatment. For example, this article published on a mdpi journal is relevant:

https://www.mdpi.com/2073-8994/1/2/215/pdf

It will be nice to see this article published as it will be an extension of the literature.

Answer: thank you for your insightful suggestion. We discussed this and added more references on the neuro-psychomotor aspects of ASD in the revised version of the manuscript.